# Structural Component Phenotypic Traits from Individual Maize Skeletonization by UAS-Based Structure-from-Motion Photogrammetry

**Monica Herrero-Huerta** [1,2,*], **Diego Gonzalez-Aguilera** [1] **and Yang Yang** [2]

1  Department of Cartographic and Land Engineering, Higher Polytechnic School of Avila, Universidad de Salamanca, Hornos Caleros 50, 05003 Avila, Spain
2  Institute for Plant Sciences, College of Agriculture, Purdue University, West Lafayette, IN 47906, USA
*  Correspondence: monicaherrero@usal.es

**Abstract:** The bottleneck in plant breeding programs is to have cost-effective high-throughput phenotyping methodologies to efficiently describe the new lines and hybrids developed. In this paper, we propose a fully automatic approach to overcome not only the individual maize extraction but also the trait quantification challenge of structural components from unmanned aerial system (UAS) imagery. The experimental setup was carried out at the Indiana Corn and Soybean Innovation Center at the Agronomy Center for Research and Education (ACRE) in West Lafayette (IN, USA). On 27 July and 3 August 2021, two flights were performed over maize trials using a custom-designed UAS platform with a Sony Alpha ILCE-7R photogrammetric sensor onboard. RGB images were processed using a standard photogrammetric pipeline based on structure from motion (SfM) to obtain a final scaled 3D point cloud of the study field. Individual plants were extracted by, first, semantically segmenting the point cloud into ground and maize using 3D deep learning. Secondly, we employed a connected component algorithm to the maize end-members. Finally, once individual plants were accurately extracted, we robustly applied a Laplacian-based contraction skeleton algorithm to compute several structural component traits from each plant. The results from phenotypic traits such as height and number of leaves show a determination coefficient ($R^2$) with on-field and digital measurements, respectively, better than 90%. Our test trial reveals the viability of extracting several phenotypic traits of individual maize using a skeletonization approach on the basis of a UAS imagery-based point cloud. As a limitation of the methodology proposed, we highlight that the lack of plant occlusions in the UAS images obtains a more complete point cloud of the plant, giving more accuracy in the extracted traits.

**Keywords:** phenotyping; unmanned aerial vehicle (UAV); photogrammetry; skeleton; deep learning



## 1. Introduction

Nowadays, climate change and environmental degradation are increasing the risk of fiber, fuel and food insecurity; cost-effective phenotyping methods are needed to meet this challenge. Traits in plants serve as features that are able to highlight the associations between genetic or physiological characteristics [1] and are imperative to plant breeding programs, biomass and yield estimations [2,3] and growth simulations [4]. Recently, phenotypic data were manually measured in the field, which is time-consuming, labor intensive and error-prone, not to mention destructive. The demand for precise agriculture and the development of close-range remote sensing technology makes image-based methods the solution to the phenotypic trait extraction challenge regarding plant physiology and structure [2], yield-related traits [3], canopy over [5] or root architecture [6,7]. Another one of the current challenges for plant phenotyping is to, accurately and with high-throughput, extract the structural components, usually composed of the root, stem, leaf, flower, fruit and seed [8]. Structural component traits are directly connected to functional phenomics,

an emerging discipline leading to an increased understanding of plant functioning by leveraging high-throughput phenotyping and data analytics [9].

Evaluating the information encoded in the shape of a plant is vital to understanding the function of plant organs [10]. A powerful shape descriptor of plant networks is the skeleton, easily computed from imaging data [11]. The skeleton opens a wide range of possibilities for quantitative phenotyping at a plant level, including describing hierarchies and branching plant networks. From the literature, there are several methods to extract the curve-skeleton from a solid, usually classified into two key types: volumetric and geometric [12]. This classification system relies on the solid's representation, depending on whether one is using an interior representation or a surface representation. Regarding volumetric approaches, they normally use a volumetric discrete representation, either a regularly partitioned voxelized representation or a discretized function demarcated in the 3D space. The potential loss of details within the solid and numerical instability due to inappropriate discretization resolution are the general disadvantages of this method [13]. On the other hand, geometric approaches directly work on the meshes or point sets. The most common used geometric methods are the Voronoi diagram [14] and medial axis [15]. Currently, Reeb graph-based methods have increased in popularity [16]. In addition, there are another group of approaches based on 3D modeling: voxel approaches and parametric surface methods. It is worth mentioning that voxel-based approaches are limited in modelling irregular surfaces.

Recently, unmanned aerial systems (UAS) have positioned themselves as a basic tool for high-throughput plant phenotyping in precision agriculture [3]. The latest advances in technology and miniaturization of their components provides additional opportunities for UAS data collection platforms. As high-resolution imaging sensors, light detection and ranging (LiDAR) has the capacity to acquire 3D measurements of plants, even in the absence of light [17,18]. This technology relies on the reflection of laser beams from the surfaces [19,20]. Currently, there are several studies using the terrestrial LiDAR to perform organ stratification (even leaf labeling) and its angles from field maize [21–24]. However, the payload reducing and cost increasing nature of LiDAR onboard UAS are the main disadvantages. On the other hand, passive imaging technologies, such as visible cameras, are lighter and less expensive. In addition, SfM (structure from motion), defined as a photogrammetric range imaging technique, offers the opportunity to acquire point clouds on the basis of images taken from various viewpoints [3]. Point clouds as three dimensional, and massive data can be used for extracting complex structural information [25]. In addition, deep learning consists of methods which can deal with object detection, classification and segmentation tasks [8], based on voxels, octree, multi-surface, multi-view and directly on point clouds. The challenge of its high cost of computing memory means these networks are mainly used in small data applications. There are some approaches using UAS imagery-based point clouds to compute basic traits such as plant height or the leaf area index in maize [26–28].

Still, methodologies to fully exploit the potential of UAS-collected data in agriculture are urgently required. In this paper, we present a novel pipeline to automatically and accurately characterize several structural component phenotypic traits in maize trails. To the best of our knowledge, the skeletonization of maize from UAS imagery-based point clouds has not been performed before. RGB images using UAS is the input of the proposed workflow to acquire a georeferenced dense point cloud of the entire study field using SfM. Topological and deep learning-based algorithms were combined to extract individual plants from the point cloud. Once a surface reconstruction process from each individual plant was achieved, the skeleton extraction algorithm was applied. Finally, we were easily able to compute structural component traits highly demanded in phenotypic tasks, comparing them with on-field and digital measurements. The paper is structured as follows: after this brief introduction, the materials, including experimental setup, data acquisition and proposed methodology are described in detail. Next, the experimental results are described,

validated and discussed. Finally, the more important conclusions reached with this study are addressed, along with future perspectives.

## 2. Materials and Methods

### 2.1. Experimental Setup and Data Acquisition

The research trial was located at the Indiana Corn and Soybean Innovation Center Manager at the Agronomy Center for Research and Education (ACRE) in West Lafayette (IN, USA) at Purdue University. Figure 1 illustrates the visualization of the workflow to follow.

## DATA ACQUISITION

1. **Ground Control Points**
2. **RGB UAS flights**
3. **Ground measures**

## METHODOLOGY

1. **Imagery-based Point Cloud**
2. **Filter**
3. **Individualization**
4. **Curve-skeleton**
5. **Phenotypic traits**

## ACCURACY ASSESSMENT

✓ **Error metrics**

**Figure 1.** Proposed workflow.

The dates of planting (DOP) were June 6 and 17, 2021. The trail was designed with an arrangement of 18 ranges and 4 rows, planting at two different densities as Figure 2 shows: approximately 14 (DOP June 17) and 18 seeds*row-1 (DOP June 6); 3 ranges and 4 rows the first density and 15 ranges and 4 rows the second one. Four GCPs (ground control points) were placed on the ground and measured using a GNSS device for georeferencing. The material of these accuracy markers was highly reflected to be easily detected in the UAS imagery dataset. The flights were carried out on 27 July (flight 1) and 3 August (flight 2), 2021 around noon solar time on sunny and no-cloud days. A Sony Alpha ILCE-7R RGB camera with a Sony 35 mm lens was the photogrammetric sensor onboard a DJI Matrice 600 Pro (M600P) platform (Gryfn, West Lafayette, IN, USA). This platform is a rotocopter UAS with onboard GPS, IMU and magnetometer and a maximum payload of 6 kg. The photogrammetric flight configuration was set up with an along- and across-track overlap of 88% and a flight altitude of 22 m. A total of 530 and 518 images from flights 1 and 2, respectively, were captured with a dimension of 7952 × 5304 pixels, given the characteristic of the photogrammetric sensor as pixel size of 4.52 μm, focal length of 35 mm and size of $35.9 \times 23.9$ mm$^2$. The sensor configuration was ISO (the International Organization of Standardization) 200, an aperture with a F-stop of f/5.6 and a fixed exposure time of 1/1250 s.

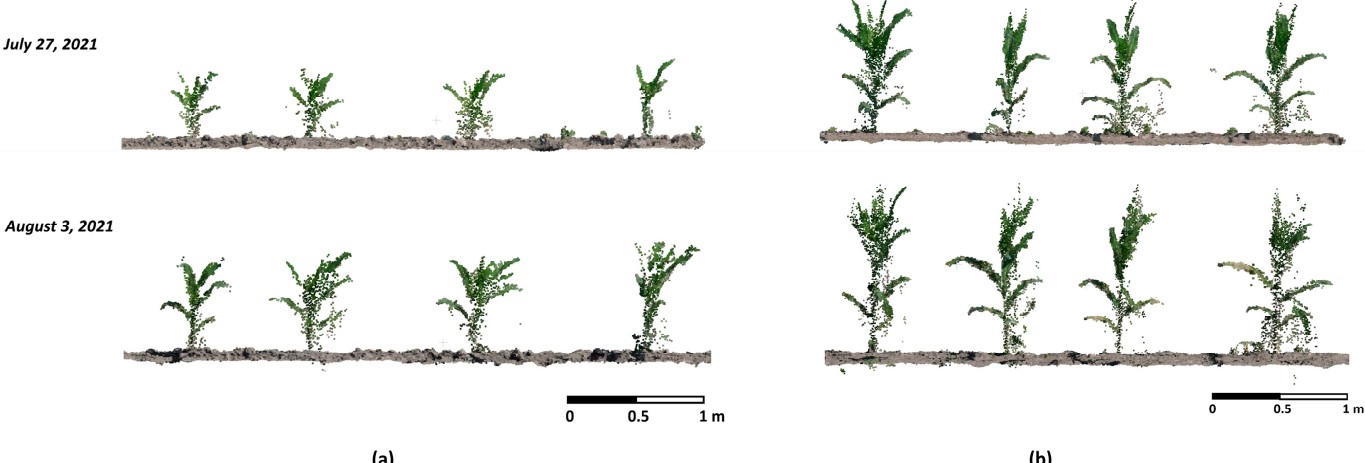

**Figure 2.** Point cloud sample processed with the imagery dataset from two flights performed (27 July and 3 August 2021) at different densities: 14 seeds*row-1 (DOP June 17) (**a**) and 18 seeds*row-1 (DOP June 6) (**b**).

As for ground measurements, stem count and plant height for the full experiment were taken at the same date as the image acquisition from UAS. Notice that before the second flight, 30% of the plants were pulled over in order to avoid occlusions from the aerial images.

### 2.2. Imagery-Based Point Cloud

Pix4Dmapper software package (Pix4D SA, Lausanne, Switzerland) was used to process aerial images, which includes camera calibration, image orientation and dense point cloud extraction. In this way, the point cloud of the study field was obtained and accurately georeferenced to the earth reference system World Geodetic System 84. However, point clouds automatically generated by SfM techniques probably englobe outlier points. To remove these points, a statistical outlier removal-based filter was applied. First, it computes the mean distance of each point to its neighbors (considering k nearest neighbors for each—k is the first parameter). Then, it rejects the points that are farther than the mean distance plus a number of times the standard deviation (second parameter). In other words, the process computes a threshold based on the Gaussian distribution of all pairwise distances in the neighborhood defined by a specific number of points (mean distance) and a number (k) to multiply the standard deviation (std. deviation), as Equation (1) shows. Points within a distance larger than the threshold are classified as outliers and removed from the point cloud [17].

$$threshold = \mu + k * \sigma \tag{1}$$

where $\mu$ is the mean distance, $\sigma$ is the standard deviation, and $k$ is a constant.

Figure 3 illustrates the low-cost photogrammetry result to 3D reconstruct a random plant from UAS imagery and the outlier removal process defined before.

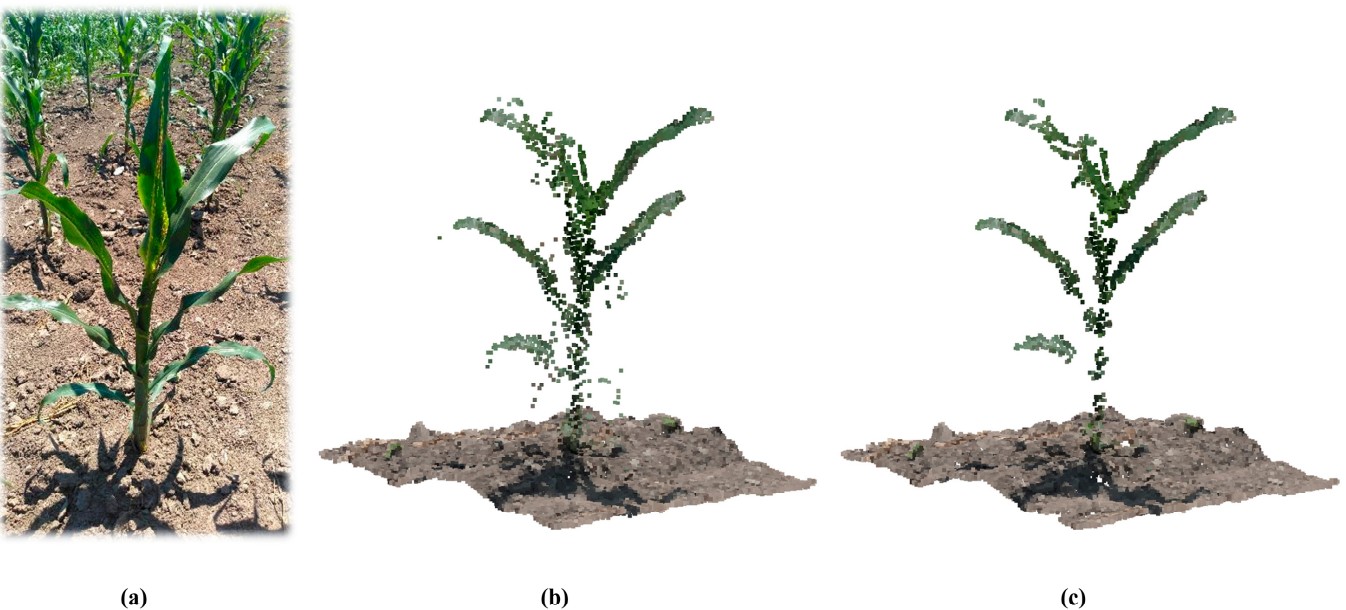

**Figure 3.** Photogrammetric 3D reconstruction of a random plant: 2D manual picture from the ground (**a**), scaled 3D point cloud from UAS imagery (**b**), clean point cloud (outlier removal) (**c**).

### 2.3. Individual Maize Extraction

A 3D deep learning unified architecture named PointNet [29] was employed to automatically perform a semantic segmentation to extract the plants from the point cloud. As the main advantage, PointNet directly runs on point clouds; that means the permutation invariance of points is not altered. Moreover, PointNet is highly robust, with little perturbation of the input points and in dealing with outliers and missing data. PointNet architecture works as follows: each point is represented by six values, its three coordinates (x, y, z) and its colors (R, G, B). The final fully connected layers of the network aggregate these optimal values into the global descriptor for the extraction. It is easy to independently apply rigid or affine transformations to each point due to the input format. Therefore, a data-dependent spatial transformer network was added, which attempted to standardize the data with the intention to further improve the results. In addition, we reduced overfitting using a data augmentation procedure that works by creating a new dataset using label-preserving transformations [30]. The first stage in the data augmentation process generates n translations in the training dataset defined by manually extracting individual maize from the point cloud. The second stage proceeds to modify the RGB intensities. For this purpose, principal component analysis was computed on the RGB value set for each training point cloud. We added multiples of the found principal components m times, with magnitudes proportional to the corresponding eigenvalues times a random variable drawn from a Gaussian with μ (mean) of zero and σ (standard deviation) of 0.1. In that way, the training set was increased by a factor of n*m. In terms of geometry and the intensity and color of the illumination, the corn plant characteristics were mainly invariant.

Once the semantic segmentation of the plants was undertaken, we extracted individual mazes by connected component labeling and setting up an octree level to define the minimum gap between two components; this means the corresponding cell size of the 3D grid for extraction [31]. This processing consists of an octree decomposition, followed by a split-and-merge procedure. First, a decomposition of a point cloud into an octree based on point density is performed. Then, the points are split within each voxel into spatially connected components. Finally, a recursive merging of components across voxels is carried out, based on a connectivity criterion until the root node is reached. As a visual example, Figure 4 displays the outputs from the steps of our pipeline to extract individual maize from the point cloud within a random plot.

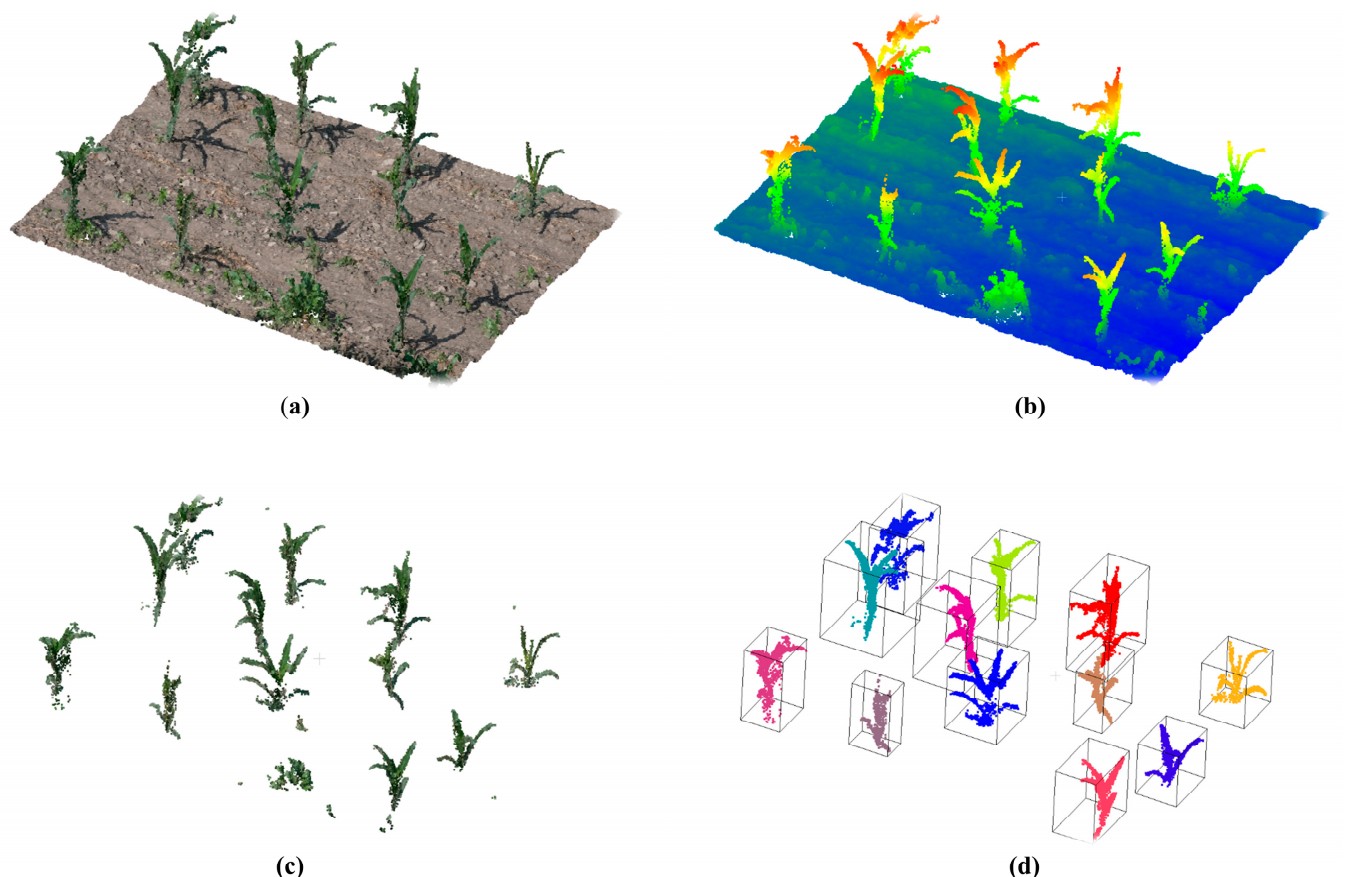

**Figure 4.** Partial and global outputs of the plant extraction pipeline within a random plot: RGB-based point cloud (**a**), height-based point cloud (**b**), vegetation-based semantic classification (**c**) and individual maize extraction and labeling (**d**).

### 2.4. Curve-Skeleton Extraction

Once the individual plants were extracted, the skeletonization process was applied to each point cloud. The skeleton structure is basically able to abstract the model volume and topological characteristics. In this case, a Laplacian-based contraction algorithm was used [13], which worked directly on the point cloud and operated on every point [32]. Advantageously, no resampled volumetric representation was required. Moreover, it was pose-insensitive and invariant to global rotation. We summarize the stages of the skeletonization process as follows: first, the mesh is contracted into a zero-volume skeletal shape, iteratively moving all the vertices along their curvature normal directions. After each iteration, all the collapsed faces from the degenerated mesh are removed until no triangles exist. During the contraction, the mesh connectivity is not altered, retaining all the key features using sufficient skeletal nodes. Lastly, the skeleton's geometric embedding is refined, moving each node to the center of mass of its local mesh region [32,33]. After these steps, we get the curve-skeleton of each individual maize.

### 2.5. Phenotypic Traits of Structural Components

The curve-skeleton is a structure that extracts the volume and topological characteristics of each individual plant represented by a point cloud and 3D line. In that way, we can easily define individual plant traits and different structural components of the plant: stem and leaves. As an individual plant phenotypic trait, we extracted the total height (difference between $z_{maximum}$ and $z_{minimum}$), crown diameter (difference between $x_{maximum}$ and $x_{minimum}$) and plant azimuth. The azimuth angle is defined as the angle between the maximum eigenvector of the plant skeleton and the north direction on the vertical

projection plane. The origin of coordinate axes was selected as the leftmost point of the plant skeleton, with a value between 0 and 180°. The stem was defined as the most vertical line. The leaves originate from stem bifurcations and have a dead-end as a topological rule. In addition, leaves must have a minimum length to be considered a proper leaf. The stem lodging was calculated by computing the orientation between medium points from the beginning and end stretch (defined by a minimum distance) of the stem skeleton; from each leaf, we mathematically computed the length based on the length of the skeleton defined as leaf and the azimuth. The leaf azimuth is defined as the angle between the maximum eigenvector of a leaf skeleton and the north direction on the vertical projection plane. The origin of the coordinate axes was selected as the connection node between the proper leaf and the stem. The value of leaf azimuth is between 0 and 360° [34]. Figure 5 shows how the traits were extracted from the skeleton.

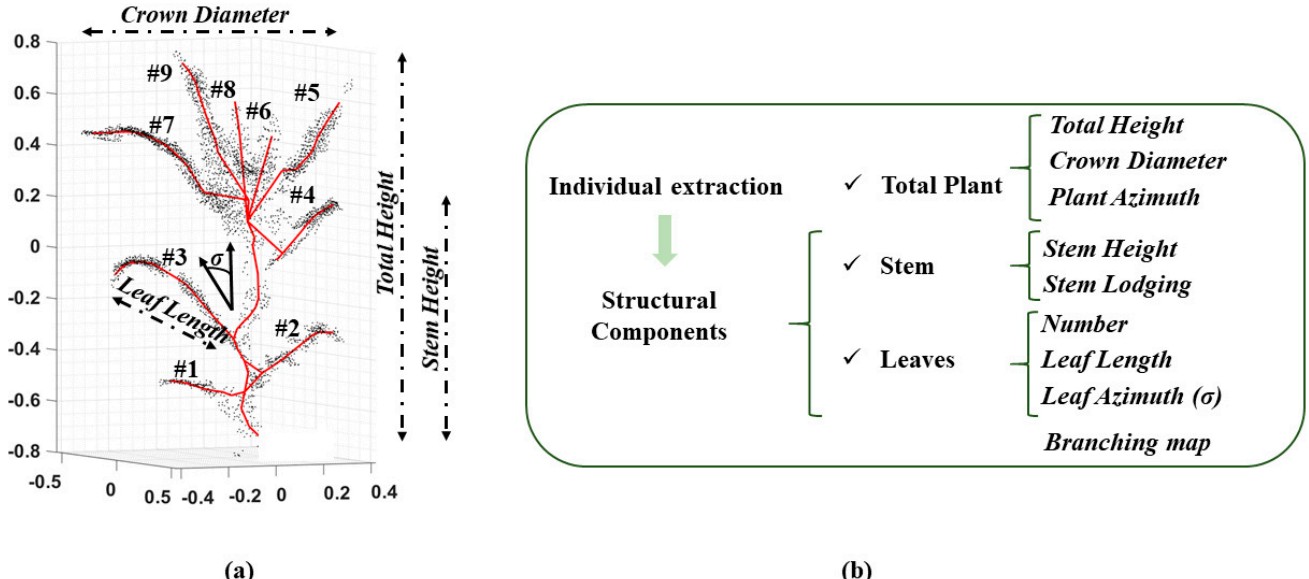

**Figure 5.** Skeleton-based structural component phenotypic traits: visual definition of the traits (**a**) and all the computed traits listed by type of the structural component (**b**).

Furthermore, this skeletal structure drives the registration process in temporal series. The registration process is critical to being able to automatically evaluate the growth of each individual plant. To register a temporal series, principal component analysis (PCA) was performed [35]. In general, the principal components are eigenvectors of the data's covariance matrix. More specifically, this statistical analysis uses the first and second moments of the curve-skeleton, resulting in three orthogonal vectors grouped on its center of gravity. The PCA summarizes the distribution of the lines along the three dimensions and models the principal directions and magnitudes of the curve-skeleton distribution around the center of gravity. Thereby, the registration of the temporal series was carried out by overlapping the principal component axes. After the registration, we can robustly monitor the growth as orientation and length variation.

*2.6. Accuracy Assessment*

The correlation between the plant height, stem count and number of leaves estimated by the skeleton and the on-field measurements or digital leaf counts was verified to evaluate the accuracy of the proposed methodology. Moreover, the rest of the skeleton algorithm-derived phenotypic traits (length and angles) were compared with manual and digital measurements from the point cloud of each individual plant. The leaf azimuth was manually measured by choosing the best suitable view direction which had the largest inclination. The determination coefficient ($R^2$), root mean square error (RMSE) and normalized root means square error (nRMSE) were calculated. The $R^2$ value was used to evaluate the coinci-

dence between the computed and the measured value. The RMSE was used to measure the deviation between both values. The nRMSE represents the degree of difference between these both values (nRMSE < 10% indicates no difference, 10% $\leq$ nRMSE < 20% denotes a small difference, 20% $\leq$ nRMSE < 30% is moderate, and nRMSE $\leq$ 30% represents a large difference) [36]. Among them, a larger $R^2$ value indicates better data fit, and smaller RMSE and nRMSE values indicate higher estimation accuracy [37]. The calculation formulas of $R^2$, RMSE and nRMSE are shown in the following Formulas (2)–(4):

$$R^2 = 1 - \frac{\sum_{i=1}^{n}\left(x_{comp}^i - \underline{x_{act}}\right)^2}{\sum_{i=1}^{n}\left(x_{act}^i - \underline{x_{act}}\right)^2} \tag{2}$$

$$RMSE = \sqrt{\frac{\sum_{i=1}^{n}\left(x_{comp}^i - x_{act}^i\right)^2}{n}} \tag{3}$$

$$nRMSE = \frac{RMSE}{\underline{x_{act}}} \tag{4}$$

where $x_{act}^i$ and $\underline{x_{act}}$ represent the actual value and the average of them, respectively (on-field measured in case of plant height and manually measured in the individual point cloud for the length and angles), $x_{comp}^i$ represents the computed value of the trait, and $n$ represents the number of samples (leaf, stem or individual plant).

Furthermore, the mean bias error (MBE), the absolute mean bias error (AMBE), the relative error (RE) and the absolute error (AE) were computed as follows (Equations (5)–(8)):

$$MBE = \frac{\sum_{i=1}^{n}\left(x_{comp}^i - x_{act}^i\right)}{n} \tag{5}$$

$$AMBE = \frac{\sum_{i=1}^{n}\left|\left(x_{comp}^i - x_{act}^i\right)\right|}{n} \tag{6}$$

$$RE = 100 * \frac{\sum_{i=1}^{n}\frac{\left(x_{comp}^i - x_{act}^i\right)}{x_{act}^i}}{n} \tag{7}$$

$$AE = 100 * \frac{\sum_{i=1}^{n}\frac{\left|\left(x_{pred}^i - x_{act}^i\right)\right|}{x_{act}^i}}{n} \tag{8}$$

In addition, the Nash and Sutcliffe index, $\eta$ is also computed (Equation (9)) and used in modelling to characterize the error related to the spatial heterogeneity:

$$\eta = 1 - \frac{\sum_{i=1}^{n}\left(x_{pred}^i - x_{act}^i\right)^2}{\sum_{i=1}^{n}\left(x_{pred}^i - \underline{x_{act}}\right)^2} \tag{9}$$

Some of these evaluation metrics have been extensively used to analysis the power of regression models [38]. Smaller values of MBE, AMBE, RE and AE and larger values of $\eta$ ($\infty < \eta \leq 1$) indicate better precision and accuracy of the prediction model.

## 3. Results

All the experimental results obtained below were run on a 3.6-GHz desktop computer with an Intel CORE I7 CPU and 32-GB RAM. We started the image processing using a Pix4D mapper software package (Pix4D SA, Lausanne, Switzerland) as a commercial solution for SfM. RGB imagery and ground control points were measured with terrestrial GPS works as inputs to finally reconstruct the study field into a scaled 3D point cloud. As an outcome, two point clouds from flights on different dates (27 July, first flight, and 3 August 2021, second flight) were computed and exactly georeferenced to EPSG 32616, WGS84 CRS. The point cloud was formed using a total of 35,983,365 points for the first flight and 34,851,008 points for the second flight, with a spatial resolution better than >24,800 points/m$^2$ in both cases. These values are valid within the limits of the study field (14 × 100 m$^2$). Due to the automated and massive character of the photogrammetric processing, an uncertainty quantity of outliers could be enclosed into these point clouds. A statistical analysis was carried out by supposing a Gaussian distribution of neighbors' distances to establish the threshold and determine outliers. The procedure has already been executed by [39]. We reached a spatial resolution better than 22,100 points/m$^2$ for both flights once the outlier detection approach was finished. A total of 257 plants for the first flight and 172 for the second flight were counted in the field, and all the plants were correctly and accurately extracted within the point clouds from both flights (30% of the plants were pulled over after the first flight and before the second in order to avoid occlusions from the aerial image). The average number of points per plant is 3968. Figure 6a represents the point clouds from the two dates and the corresponding individual plant extraction in top view. In Figure 6b, a zoomed window is shown with a 3D view. Figure 6c illustrates the growth of the individual plants within this zoomed area between the two dates precisely quantified in meters. We registered the point cloud of each individual plant from these two dates by computing PCA from the skeleton and overlapping the principal component axes. In this particular case, the maximum growth is 0.41 m. In addition, we can calculate the average maximum growth per plant, which is 0.22 m, and we can point out that the growth is always bigger at the upper part of the corn between these two dates.

Next, once the individual plants were extracted, the skeleton was computed from each point cloud using a Laplacian-based contraction algorithm, as Section 2.4 explains. Figure 7 graphically shows, in 3D, the individual point cloud in black overlapping the skeleton in red of the 16 plant cases: the maximum and minimum height, crown diameter, number of points and grown increment from the two flights (27 July 2011 and 3 August 2011). It is worth mentioning that the axes are in relative values. Below, Table 1 shows the plant data location and traits from the plant samples: the number of points, UTM coordinate of the point cloud center, and dimension of the bounding box from each individual point cloud, as well as traits computed by the individual point cloud skeletonization, such as the number of leaves, plant height, crow diameter, plant azimuth, lodging calculated as stem azimuth, stem height, mean leaf azimuth and mean leaf length.

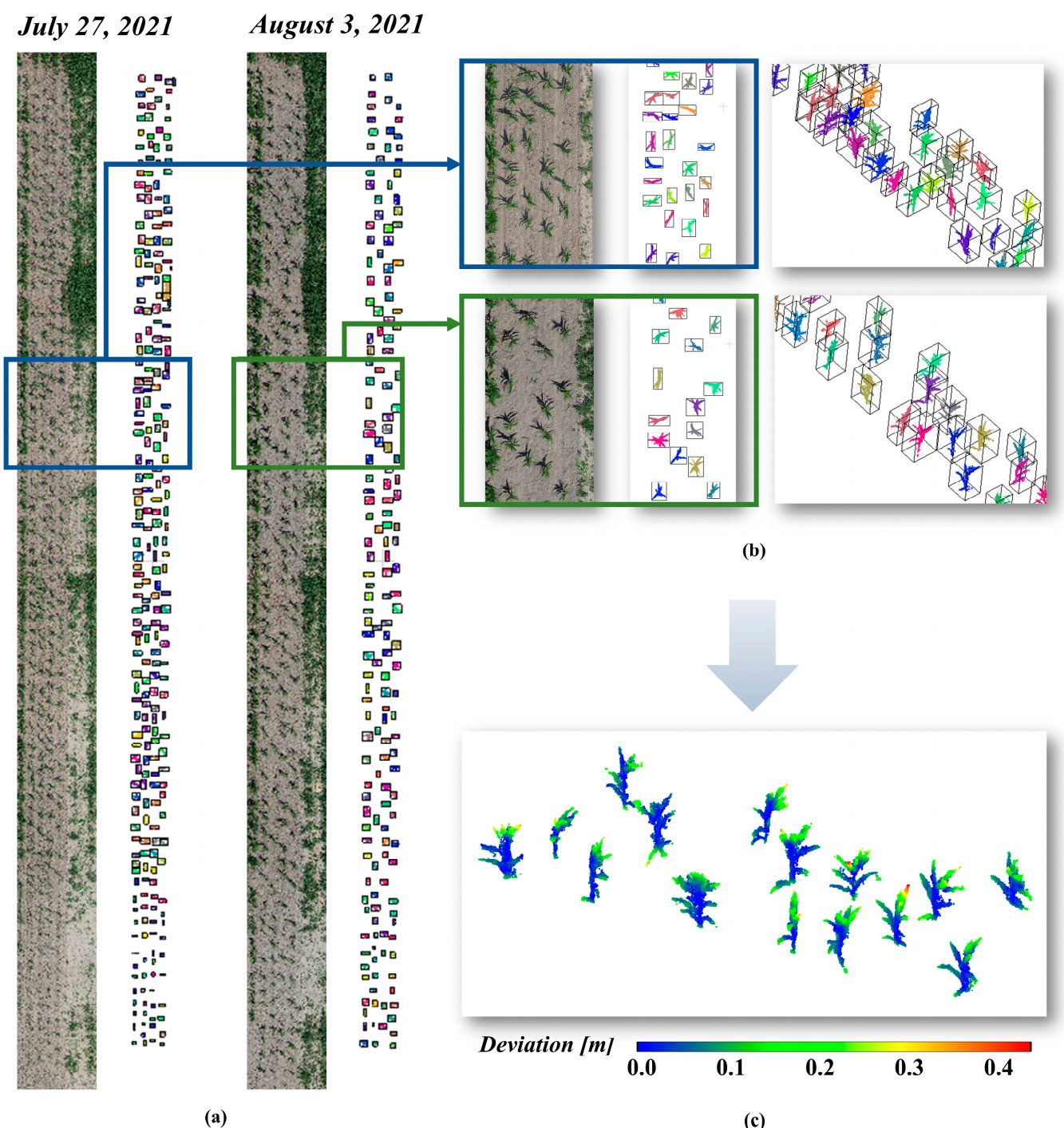

**Figure 6.** Point clouds from the two dates and individual plant extraction in different colors surrounded by a bounding box (**a**); zoomed window of a random area with a 3D view of the extracted plants (**b**); plant growth within this zoomed area from the two dates quantified in meters (**c**).

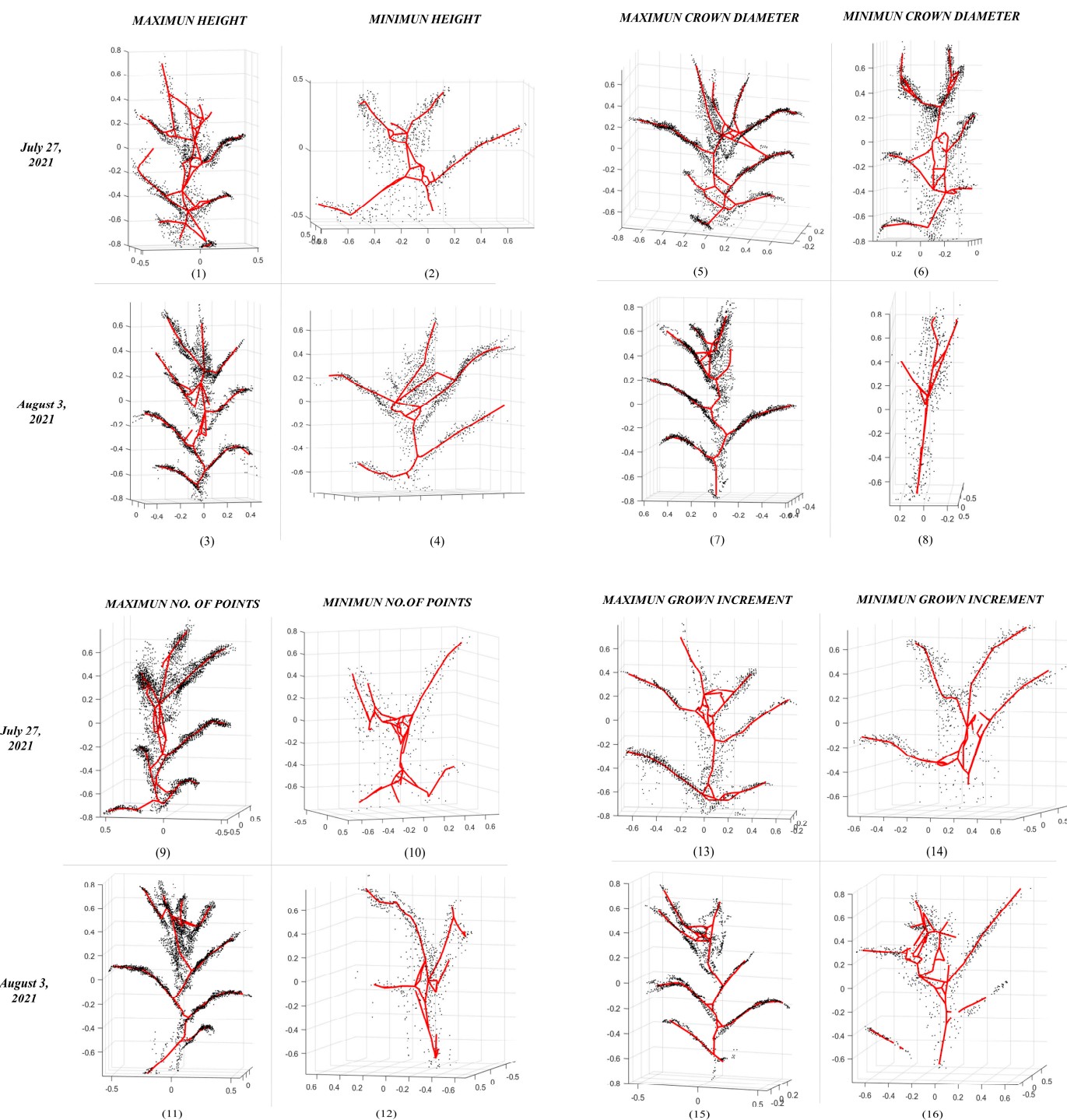

**Figure 7.** The 3D skeleton extraction in red overlapped the individual point cloud in black of 16 plant samples: maximum and minimum height, crown diameter, number of points and grown increment from the two flights (27 July 2011 and 3 August 2011).

**Table 1.** Plant data and traits from the 16 plant samples: number of points, UTM coordinate of the point cloud center and dimension of the bounding box from the point cloud; traits computed by the individual point cloud skeletonization, such as number of leaves, plant height, crow diameter, plant azimuth, lodging calculated as stem azimuth, stem height, mean leaf azimuth (LA) and mean leaf length (LL).

| Plant ID | Point Cloud | | | | | | | Skeleton | | | | | | | |
| | | UTM Coord. Center | | | Bounding Box (m) | | | | | | Trait Extraction | | | | |
| | N°. Points | AX+500,000 | AY+4,480,000 | AZ+0 | X | Y | Z | #Leaves | Height (m) | Crow Diam. (m) | Plant Azimuth (°) | Lodging (°) | Stem Height (m) | Mean LA (°) | Mean LL (m) |
|---|---|---|---|---|---|---|---|---|---|---|---|---|---|---|---|
| 1 | 2429 | 224.21 | 202.40 | 181.93 | 0.66 | 0.30 | 1.04 | 9 | 0.97 | 0.69 | 339.4 | 351.0 | 0.58 | 344.4 | 0.21 |
| 2 | 956 | 225.36 | 156.35 | 180.90 | 0.38 | 0.26 | 0.25 | 4 | 0.22 | 0.37 | 338.2 | 341.1 | 0.16 | 340.7 | 0.13 |
| 3 | 3481 | 223.47 | 205.84 | 182.20 | 0.43 | 0.82 | 1.32 | 8 | 1.29 | 0.78 | 1.1 | 3.2 | 0.87 | 1.4 | 0.34 |
| 4 | 1729 | 223.11 | 156.49 | 181.19 | 0.58 | 0.34 | 0.32 | 5 | 0.29 | 0.52 | 27.8 | 19.4 | 0.24 | 24.3 | 0.20 |
| 5 | 3929 | 223.33 | 190.44 | 181.87 | 0.90 | 0.31 | 0.83 | 8 | 0.81 | 0.88 | 6.7 | 8.4 | 0.56 | 8.3 | 0.24 |
| 6 | 2259 | 225.88 | 221.57 | 181.49 | 0.16 | 0.23 | 0.81 | 7 | 0.79 | 0.24 | 358.1 | 5.5 | 0.61 | 1.7 | 0.12 |
| 7 | 3891 | 223.3 | 172.99 | 181.98 | 0.89 | 0.64 | 1.15 | 7 | 1.12 | 0.82 | 3.9 | 2.5 | 0.89 | 2.7 | 0.30 |
| 8 | 740 | 223.79 | 149.17 | 180.86 | 0.139 | 0.31 | 0.39 | 3 | 0.36 | 0.29 | 19.8 | 23.3 | 0.16 | 22.0 | 0.07 |
| 9 | 4641 | 224.96 | 199.35 | 181.90 | 0.59 | 0.59 | 0.87 | 7 | 0.85 | 0.57 | 18.8 | 8.1 | 0.60 | 15.6 | 0.21 |
| 10 | 369 | 223.85 | 149.24 | 180.68 | 0.44 | 0.40 | 0.26 | 5 | 0.23 | 0.47 | 9.4 | 12.4 | 0.12 | 10.8 | 0.06 |
| 11 | 5045 | 223.44 | 197.97 | 182.17 | 0.88 | 0.58 | 1.22 | 9 | 1.20 | 0.81 | 357.6 | 1.2 | 0.89 | 259.6 | 0.32 |
| 12 | 547 | 223.22 | 158.67 | 181.20 | 0.36 | 0.42 | 0.40 | 4 | 0.35 | 0.40 | 345.6 | 354.8 | 0.13 | 347.2 | 0.10 |
| 13 | 2115 | 225.82 | 212.08 | 181.68 | 0.86 | 0.45 | 1.00 | 6 | 0.97 | 0.79 | 17 | 4.0 | 0.49 | 4.9 | 0.23 |
| 14 | 947 | 223.87 | 153.23 | 180.80 | 0.40 | 0.43 | 0.45 | 4 | 0.39 | 0.41 | 2.9 | 6.2 | 0.16 | 5.8 | 0.15 |
| 15 | 1636 | 225.72 | 212.19 | 181.68 | 0.74 | 0.23 | 0.82 | 8 | 0.79 | 0.68 | 349.2 | 356.1 | 0.50 | 355.6 | 0.38 |
| 16 | 847 | 223.85 | 153.22 | 180.84 | 0.30 | 0.34 | 0.56 | 8 | 0.51 | 0.38 | 6.1 | 4.2 | 0.14 | 6.7 | 0.11 |

## 4. Discussion

In this section, the results are discussed and validated. The stem counts measured in the field with GPS were exactly the same as the digitally taken stem counts for both flights: 257 plants for the first flight and 172 for the second one. The individual height of each plant was also measured in the field using a tape with centimetric precision. Comparing this on field-measurement with the digital height computed from the point cloud of each extracted plant, an R2 of more than 0.99 was achieved. No outliers were detected in this regression, guaranteeing accurate and precise height results. From Table 1, it is remarkable that the plants with a greater number of leaves are the ones with the maximum plant height and a greater number of points. It seems reasonable because when the point density is higher, the plant has more detail to distinguish the leaves, and higher plants have more chance to have leaves. On the other hand, the plants with less recognizable leaves coincide with the minimum crown diameter plants. Another bit of information we can extract is that the more vertical plants are the highest ones, while the more inclined plants (lodging) coincide with the minimum crown diameter one. The following table, Table 2, illustrates statistics values from the computed traits of all the individual plant point clouds from both flights (27 July and 3 August 2011): mean, standard deviation (Std), median, normalized median absolute deviation (NMAD) (Equation (10)) and square root of the biweight midvariance (BwMv) (Equation (11)). It is worth mentioning that for the computation of the table, the outliers were discarded according to the studentized residuals for a significance level of 0.05 with two-tail distribution.

$$NMAD = 1.4826 \cdot MAD \tag{10}$$

$$BwMv = \frac{n \sum_{i=1}^{n} a_i (x_i - m)^2 (1 - U_i^2)^4}{\left( \sum_{i=1}^{n} a_i (1 - U_i^2)(1 - 5U_i^2) \right)^2} \tag{11}$$

$$a_i = \{1, if\,|U_i| < 1\ 0, if\,|U_i| \geq 1 \tag{12}$$

$$U = \frac{x_i - m}{9MAD} \tag{13}$$

being the median absolute deviation (MAD) and the median (m) of the absolute deviations from the data's median (mx).

**Table 2.** Statistics of the computed traits (mean, Std, median, NMAD and BwMv) and error metrics of all plants from both flights at 95% confidence interval (MBE, AMBE, RMSE, NMAD, RE, AE and $\eta$).

| | #Leaf | Height (cm) | Crown Diam. (cm) | Azimuth (º) | Lodging (º) | $H_{stem}$ (cm) | Mean LA (º) | Mean LL (cm) |
|---|---|---|---|---|---|---|---|---|
| Mean | 5.98 | 70.16 | 54.66 | 1.18 | 4.56 | 42.76 | −4.94 | 19.19 |
| Std | 1.40 | 34.84 | 21.43 | 15.61 | 8.48 | 28.12 | 26.82 | 9.61 |
| Median | 7 | 79.81 | 54.91 | 1.71 | 4.66 | 51.57 | 3.54 | 20.55 |
| NMAD | 2.43 | 44.84 | 28.45 | 14.45 | 13.2 | 35.80 | 23.13 | 11.44 |
| BwMv | 0.25 | 82.66 | 31.43 | 12.22 | 4.82 | 53.80 | 24.65 | 6.22 |
| $R^2$ (%) | 90.9 | 99.8 | 99.7 | 99.8 | 99.9 | 99.4 | 99.7 | 68.8 |
| RMSE | 0.661 | 1.769 | 1.137 | 8.456 | 4.650 | 2.341 | 11.054 | 8.231 |
| nRMSE (%) | 10.5 | 2.5 | 2.0 | 6.1 | 4.9 | 5.2 | 6.1 | 32.7 |
| MBE | 0.063 | −0.431 | −0.150 | −3.375 | −2.125 | −0.544 | −3.563 | −5.000 |
| AMBE | 0.438 | 1.244 | 0.888 | 6.000 | 3.875 | 1.906 | 10.063 | 5.613 |
| RE | 0.781 | −0.026 | −0.052 | −0.429 | −0.122 | −0.333 | −0.294 | −2.780 |
| AE | 0.781 | 0.026 | 0.104 | 0.429 | 0.122 | 0.333 | 0.294 | 14.053 |
| $\eta$ | 0.879 | 0.997 | 0.997 | 0.997 | 0.999 | 0.993 | 0.996 | 0.267 |

Analyzing the computed traits, the plant height and the stem height have more dispersed values (in this position), as the larger values of NMAD and BwMv show. With regard to errors, the determination coefficient is superior to 0.9 for all the traits, except for the mean leaf length. The algorithm fails to recognize the short leaves, and it, therefore, concludes that the mean of the computed leaf length is much larger. This is the same reason

why the normalized root mean square error and the Nash and Sutcliffe index get the worst score in this computed trait. The rest of the errors show no difference between the actual and computed values at all.

### 5. Conclusions

As this study highlights, skeletons are powerful descriptors for analyzing plant networks by defining structural components and computing several phenotypic traits. Moreover, close-range platforms together with novel deep learning networks show a powerful combination to extract individual maize plants. Therefore, the approach proposed here is pretty rapid, accurate and cost-effective. It is worth mentioning that particular attention has been paid to the spatial resolution and completeness of the computed point cloud to effectively run our approach. These aspects are directly related to the plant spacing, which could generate shadows and to the variables coming from the flight (overlap, altitude and flight direction) to get a dense point cloud. In this study, the image acquisition strategy was only from nadir. However, oblique images will improve the completeness of the plant point cloud. Future analyses are needed to be able to apply our pipeline to different plant species and phenotypic growth stages, as well as to investigate the influence of environmental factors such as soil properties and light conditions. In addition, several platforms for high-throughput phenotyping, even terrestrial platforms or LiDAR-collected point clouds, are intended to be tested.

**Author Contributions:** M.H.-H. conceived the idea, developed the data analysis pipelines and software, performed the data analysis and visualization and wrote the manuscript; D.G.-A. supported the research and edited the manuscript; Y.Y. supervised the research. All authors have read and agreed to the published version of the manuscript.

**Funding:** MHH was supported by the Spanish Government under Maria Zambrano (Requalification of the Spanish university system for 2021–2023). This research was also funded by the European project H2020 CHAMELEON: A Holistic Approach to Sustainable, Digital EU Agriculture, Forestry, Livestock and Rural Development based on Reconfigurable Aerial Enablers and Edge Artificial Intelligence-on-Demand Systems. Ref: 101060529, Call: HORIZON-CL6-2021-GOVERNANCE-01-21.

**Institutional Review Board Statement:** Not applicable.

**Informed Consent Statement:** Not applicable.

**Data Availability Statement:** The data analyzed during the current study (the point cloud of the area of interest on the two dates and the point cloud of each individual plant and its skeleton) can be found as supplementary data at https://drive.google.com/file/d/1Yp-5ujfXRvtNqqmZ4xr-uRGKvXKJjpYM/view?usp=share_link (accessed on 9 January 2020).

**Acknowledgments:** The authors would like to thank the Institute for Plant Sciences (College of Agriculture), Jason Adams, Brian Dilkes and his lab and all at Purdue University (IN, USA) for their collaboration during the experimental phase of this research.

**Conflicts of Interest:** The authors declare no conflict of interest. The funders had no role in the design of the study; in the collection, analyses, or interpretation of data; in the writing of the manuscript; or in the decision to publish the results.

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
