# Peer review of "Structural Component Phenotypic Traits from Individual Maize Skeletonization by UAS-Based Structure-from-Motion Photogrammetry"

_drones, doi:10.3390/drones7020108_

Round 1
Reviewer 1 Report
The overall writing of this paper is relatively mature, but the innovation points are not outstanding. The point cloud segmentation of maize monocots is now a very mature technology, so the innovation of the paper is lacking.
Author Response
We would like to thanks to the reviewers and to the editor. The current version is now a better version.
All the comments are answered in the 'response to reviewers' document.
Thanks

Reviewer 2 Report
The study aimed at the topical problem of agricultural studies by UAV technologies. The paper presented a workflow for the automatic and accurate characterization of several structural component phenotypic traits in maize trails.
There are some remarks to the paper.
1. Be careful. You are treating R2 wrong as a correlation coefficient. In fact, this is a determination coefficient and describes the amount of explained model variability (response of dependent variable explained by regression model). In contrast, the correlation coefficient represents the relationship level between variables.
2. Despite the well-described workflow, I would recommend visually presenting the workflow in the form of a diagram to grasp the whole process easily.
3. Fig. 6 quality must be improved.
4. Please revise the very beginning of section 4. It seems that you forgot to get rid of the MDPI recommendations for the section content.
Author Response

(The authors gave the same response as above.)

Reviewer 3 Report
Originality:
- The article makes a valuable contribution in the field of agricultural monitoring using UAV. Other research exists in the area of maize monitoring.
Scientific Quality:
- Well written paper. The content of the article shows the scientific approach. Tthe proposed method is described coherently. The limits of the proposed methodology could be mentioned.
Relevance to the Field(s) of this Journal:
- The research problem presented here is very interesting and fits well to the fields covered by this journal.
General Comment:
- Is the onboard equipment (line 112) only capable of receiving GPS signals or even other GNSS signals (e.g. GLONASS, Galileo, etc.)?
- There are multiple errors in equation (1): “threshold” and “mean distance” are written incorrectly. In general, it is recommended to use the common notations. In this case use µ for mean and σ for standard deviation.
- Line 168: “standard deviation of 0.1”
- It is not quite clear how to understand the statement from line 287,288, 289. Were there a total of 257 plants (= ground truth) on the 1st date and all were detected or were 257 of a total of X detected? In addition, it is noticeable that one week later significantly fewer plants were detected. Here it would be useful to indicate possible reasons why fewer were detected, for example: Did the number of plants decrease within the 7 days due to external environmental influences (thunderstorms, hail, etc.)? Are the corn plants more difficult to separate from each other due to advanced growth? Do different environmental influences at the time of the two flight tests indicate the changed detection performance?
Abstract:
- The abstract includes the purpose of the research. It also describes the experimental setup and methodology. Results are discussed, limitations of the presented methodology could be added. Keywords are well chosen.
Introduction:
- Introduction provides background information and describes the research problem as well as research objectives.
- Is the term “remote sensing” (line 40) in the right context here? Doesn't that fit more with large-scale observation of agricultural land from a satellite-based perspective, rather than measurement methods flown at 22m?
Literature Review:
- The literature used is very up to date and adequate for the journal article. A little more could be said about the state of the art of the subject with reference to similar research activities and how one's own research differs from them.
Methodology:
- The methodology used is described in detail. The proposed solutions and tools used for the research problem are up-to-date and the experimental setup is thoroughly explained.
Results:
- The presented method allows determination of growth between two dates and allows conclusions about growth differences in different areas of the maize field. In addition, statements can be made about structural components of maize plants, which is a valuable contribution to monitoring maize acreage. Meaningful state-of-the-art evaluation metrics are used to evaluate the results.
Conclusions:
- The conclusion turns out to be quite short. In addition to the investigation of further plant types, future work would also be interesting to investigate the influence of environmental factors (soil properties, light conditions) as well as the influence of plant spacing or the altitude flown.
References / Bibliography:
- Citation style conforms to journal standards. As already mentioned, there is other similar research work concerning maize monitoring, which is not referred to. Examples are: “UAV-based imaging platform for monitoring maize growth throughout development” or “Estimating leaf area index of maize using UAV-based digital imagery and machine learning methods”. In future publications, a state of the art chapter could refer more to research with similar research questions and distinguish one's own work from them.
Figures:
- In Figure 3, the section from an original recording would be additionally interesting to be able to compare the results more effectively.
- The image quality of Figure 6 needs to be improved. For example, the axis labels and the title of the plots are barely legible.
Tables:
- Table 2.: “Crown” written incorrectly
Others:
- Starting on page 10, the page references are inconsistent and sometimes missing at all.
Author Response

(The authors gave the same response as above.)
